# The carbon footprint of surgical operations: 2023–2025 systematic review update

Anna Savio[1]*, Beatrice Marchi[1], Andrea Roletto[1], Pierangelo Guizzi[2],
Giuseppe Milano[3], Lucio Enrico Zavanella[1], Simone Zanoni[4]

**1** Department of Mechanical and Industrial Engineering, Università degli Studi di Brescia, Brescia, Italy,
**2** Department of Orthopaedic Surgery, Ospedale di Gardone Val Trompia, Brescia, Italy, **3** Department
of Medical and Surgical Specialties, Radiological Sciences, and Public Health, Università degli Studi di
Brescia, Brescia, Italy, **4** Department of Civil, Environmental, Architectural Engineering and Mathematics,
Università degli Studi di Brescia, Brescia, Italy

* anna.savio@unibs.it

journal.pone.0349415

Universidad de Jaen, SPAIN

**Peer Review History:** PLOS recognizes the
benefits of transparency in the peer review
process; therefore, we enable the publication
of all of the content of peer review and
author responses alongside final, published
articles. The editorial history of this article is
available here: https://doi.org/10.1371/journal.
pone.0349415

## Abstract

Surgical care significantly contributes to greenhouse gas emissions, with operating
rooms consuming three to six times more energy than other departments relying
heavily on single-use materials and anaesthetic gases. We conducted a systematic
review of literature from 2023 to 2025 following PRISMA guidelines. The PubMed,
Scopus and Healthcare LCA databases were searched, and eligibility criteria were
applied. A total of 26 studies met inclusion criteria. Data were extracted on each
study's scope and system boundaries, reported emission values, identified carbon
hotspots, and any mitigation strategies evaluated. A structured quality assessment
was conducted to evaluate the reliability of the included studies and to identify poten-
tial sources of bias. The included studies demonstrated wide variability in reported
carbon footprints of surgery, ranging from under 5 $kgCO_2e$ for minor procedures to
over over 400 $kgCO_2e$ for complex single-stage operations and approximately 1,000
$kgCO_2e$ for whole multistage patient pathways. This systematic review underscores
that surgical operations could have a significant carbon footprint, with emissions
hotspots concentrated in consumable materials and energy consumption. It also
reveals substantial variability and methodological heterogeneity in how surgical car-
bon footprints are calculated, pointing to the urgent need for standardized life-cycle
based frameworks in surgical settings. Establishing common standards will enable
more reliable benchmarking of surgical emissions and better comparisons across
studies.

## Introduction

Climate change poses an urgent threat to global health, and the healthcare
sector is a significant contributor to greenhouse gas (GHG) emissions. World-
wide, healthcare is responsible for roughly 4–5% of total GHG emissions. In

**Data availability statement:** All relevant data are within the manuscript and its Supporting Information files.

**Funding:** The author(s) received no specific funding for this work.

**Competing interests:** The authors have declared that no competing interests exist.

**Abbreviations:** $CH_4$, Methane; $CO_2$e, Carbon dioxide equivalent; CT, Computed tomography; GHG, Greenhouse gas; GWP, Global warming potential; HVAC, Heating, Ventilation and Air Conditioning; ISO, International Organization for Standardization; kWh, Kilowatt-hours; LCA, Life-cycle assessment; MRI, Magnetic resonance imaging; $N_2O$, Nitrous oxide; OR, Operating room.

high-income countries, this share can be even greater (e.g., 8% in the US and 5.4% in the UK) [1–3]. Within healthcare, hospitals, and specifically surgical services, are known to be particularly energy and resource intensive. Operating theatres have been reported to be three to six times more energy-intense than the hospital, making them major hotspots for emissions [4]. They also generate disproportionately large amounts of waste, 20–70% of a hospital's waste comes from operating rooms. As a result, surgery's environmental footprint has come under increasing scrutiny from both healthcare professionals and policymakers aiming to achieve carbon reduction targets, such as the UK National Health Service (NHS) commitment to achieve its interim target of an 80% reduction in direct emission by 2032 [5].

Despite this importance, systematic assessments of the carbon footprint of surgical operations only began to appear in the literature over the past decade. A landmark systematic review in 2020 was the first to compile studies of surgical carbon footprints [6]. Searching over 50 years of records, the study identified just 8 scientific articles, all published after 2011, that quantified the carbon footprint of any surgical procedure. Those studies reported a wide variation in per-procedure emissions, from about 6 $kgCO_2$e up to 814 $kgCO_2$e for a single operation. The largest values were associated with complex surgeries (e.g., some robotic or cardiac procedures). Across the board, medical devices and consumable supplies were identified as the biggest contributors to surgical emissions. The authors noted marked methodological heterogeneity and called for standardized reporting to enable reliable comparison.

An updated systematic review in 2023 sought to incorporate new studies that had emerged since 2020 [7]. This review included 7 studies in total. It found a slightly narrower range of carbon footprints, from 28.5 $kgCO_2$e to 505.1 $kgCO_2$e per operation, across those studies. Single-use devices and consumables again emerged as the dominant emission source, and the authors reiterated the need for international standards in surgical carbon footprinting.

Since early 2023, however, interest in the environmental sustainability of surgery has grown rapidly, and many new studies have been published across various surgical specialties. This systematic review aims to build upon and update the prior reviews with the latest evidence. We incorporate 26 new studies that quantify the carbon footprint of surgical operations, providing the most up-to-date and comprehensive synthesis of data in this emerging field. In doing so, we expand the scope to cover a wider range of procedures, including orthopaedic surgeries, obstetric and gynaecologic procedures, otolaryngology surgeries, urologic oncology, plastic surgery, cardiothoracic surgery, and more, and new geographical settings.

The objectives of this review are to:

a) Quantitatively summarize the carbon footprints reported for different surgical procedures and specialties in recent literature, including typical ranges and outliers. We compare footprints across procedure types, and, where possible, across different countries/health systems.

b) Identify common emissions hotspots in surgical pathways, such as the contribution of single-use consumables, energy use, anaesthetic gases, patient/staff travel, sterilization processes, and waste disposal, as reported by these studies.

c) Highlight proposed mitigation strategies and interventions that have been evaluated or suggested to reduce surgical carbon footprints. This includes measures like switching anaesthetic techniques, using reusable instrumentation, waste reduction and recycling initiatives, optimizing OR utilization and energy efficiency, among others.

d) Assess methodological variability among studies, including differences in life-cycle assessment (LCA) boundaries, functional units, GHG scopes, and data sources for emission factors. We discuss how these differences impact results and interpretability.

By addressing these aims, this review provides a timely and comprehensive overview for surgeons and health system leaders interested in understanding and reducing the environmental impact of surgical care. It also serves to update prior reviews by incorporating the surge of recent evidence, identifying knowledge gaps and research needs for the field of sustainable surgery. Ultimately, improving the sustainability of surgery aligns with the broader healthcare mission to "do no harm", recognizing that protecting planetary health is integral to protecting human health.

## Methods

### Protocol and reporting

This review was designed and reported in accordance with the PRISMA 2020 guidelines for systematic reviews and meta-analyses [8] and registered with the International Prospective Register of Systematic Reviews (PROSPERO, ID CRD420251175778). The PRISMA flow diagram illustrating the study identification, screening, eligibility, and inclusion process is shown in Fig 1. The PRISMA checklist is provided in the Supporting Information (S1 File).

### Eligibility criteria

We sought to include studies that quantitatively evaluated the carbon footprint of surgical operations. Specifically, eligible publications had to report a carbon footprint calculation or life-cycle assessment (LCA) for one or more surgical procedures or surgical care pathways. We included original research articles, both prospective and retrospective analyses were eligible, as well as modelling studies, case studies, and comparative studies (e.g., comparing techniques or scenarios) as long as a carbon footprint or $CO_2e$ emission was quantified.
We excluded:

- Non-original contributions (editorials, letters);

- Systematic reviews/meta-analyses (used only as secondary sources);

- Articles that only measured waste or energy consumption without translating them into $CO_2e$;

- Studies focused exclusively on a single element (e.g., anaesthetic gasses) without the overall context.

Some partial studies (e.g., only on waste) were included if they reported $CO_2e$ estimates, in order to ensure comparative breadth. No restrictions were applied regarding surgical specialty, type of intervention, or country.

No language restrictions were applied, ensuring that studies were identified and screened regardless of the language of publication and minimizing the risk of language-related selection bias. When full texts were not available online, the corresponding authors were contacted; studies were excluded if no response was received. The publication date range was January 1, 2023, through September 1, 2025.

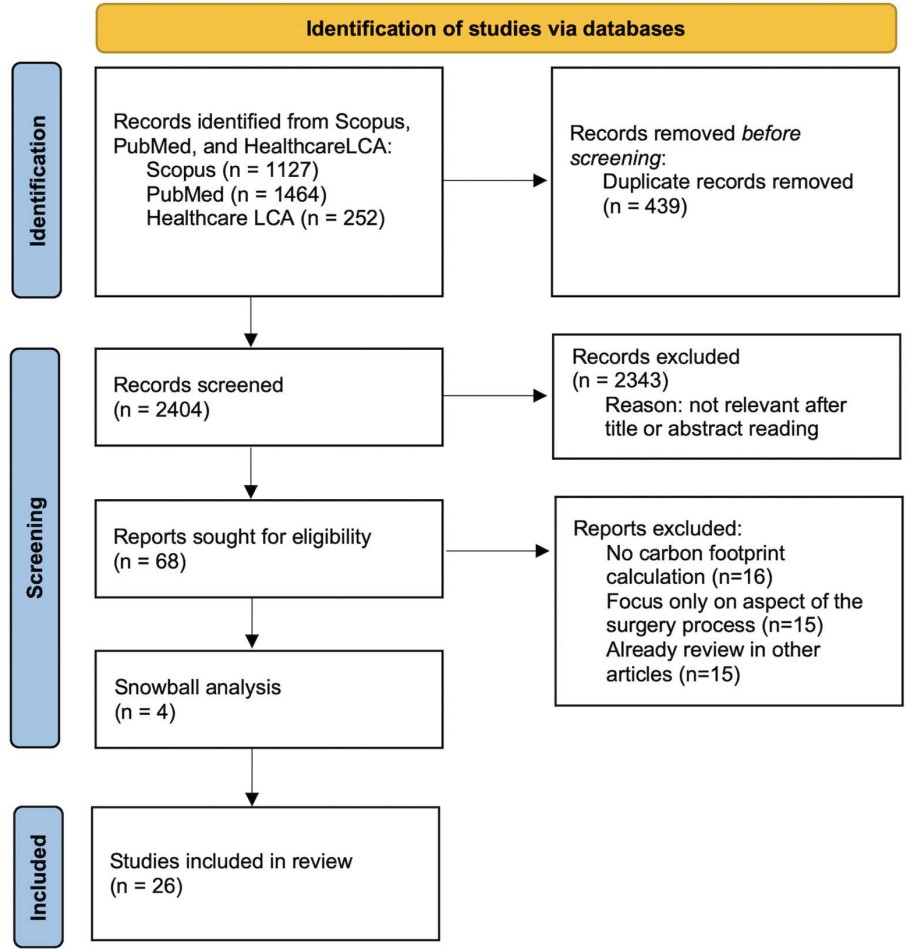

**Fig 1. PRISMA flow chart of study selection process.**

## Information sources and search strategy

We performed systematic searches in multiple databases: PubMed, Scopus, and a specialized database for healthcare life-cycle assessments, the Healthcare LCA database [9]. Our search strategy combined terms related to environmental impact and sustainability, healthcare settings, and surgical, clinical, and diagnostic activities. The complete search strings for all databases are reported in the Supporting Information (S2 File). The final database searches were conducted in September 2025. We also examined the reference lists of relevant articles (including the prior reviews and any new studies found) for snowball sampling of potentially missed studies. Using Google Scholar, we identified a few additional records that did not appear in the database search but met inclusion criteria. These were screened and added if eligible. As shown in Fig 1, 4 studies were included via this snowball method, supplementing the database search results. All references were imported into a reference manager, and duplicates were removed (both automatic deduplication and manual checking).

## Selection process

Two reviewers independently assessed titles and abstracts (n = 3,293). With an inclusive approach, every uncertain citation has been advanced to the full text. 68 full articles were examined and 46 were excluded for:

- No carbon footprint calculation (16 studies): The article discussed environmental issues in surgery but did not actually quantify $CO_2$e emissions;

- Focused only on a sub-component (15 studies): The analysis was restricted to a single aspect such as anaesthetic gas usage or sterilization energy and did not consider overall surgical footprint;

- Already covered by other included studies or reviews (15 studies): this included, for instance, earlier reports that were superseded by updated analyses from the same group, or cases where the data were part of a larger study that we included. Systematic reviews were also excluded in this category.

Any disagreements in inclusion were resolved through discussion, and if needed, consultation with a third reviewer. Ultimately, we reached consensus to include 26 studies in the qualitative synthesis.

The PRISMA flow (Fig 1) summarizes this selection: out of >4,200 records identified (including 252 from the specialized LCA source), 3,293 unique records were screened, 3,228 were excluded at title/abstract stage, 68 full texts were assessed, 4 were identified with a snowball analysis and 26 met criteria and were included.

### Data extraction and synthesis

A data extraction form was developed and pilot-tested on a few initial papers to ensure consistency with the approach used in prior reviews. For each included study, we extracted key characteristics and results, including:

a) Study reference (author, year) and geographical setting;

b) Clinical context and surgical procedure(s) assessed, including any comparative techniques or scenarios within the study;

c) Methodology approach: type of carbon accounting or LCA, standards or guidelines referenced (e.g., ISO 14040, GHG Protocol, etc.), system boundaries, study design (prospective/retrospective), and the sample size;

d) Functional unit and analytical scope, including time boundaries and emission sources included or excluded;

e) Greenhouse gases accounted for and sources of emission factors (e.g., Ecoinvent, national datasets);

f) Main carbon footprint outcomes (kgCO$_2$e per functional unit), including all reported comparators and summary statistics;

g) Key findings related to mitigation opportunities or emissions drivers;

h) Notes on study design, assumptions, and limitations relevant to interpretation.

The compiled extraction table was used as the basis for the results synthesis and for constructing Table 1 and Table 2 below. The results of the quality assessment are reported in S1 Table, while S2 Table summarizes the type of data collected, study scope, and emission factors considered. Stated boundary exclusions, assumptions made during data collection, and other study limitations are detailed in S3 Table.

## Results

### Overview of included studies

We included 26 studies published from 2023 to September 2025, reflecting the surge in research on surgical environmental impact in this period.

Table 1 presents a summary of each included study, organized by year and first author, with information on country, surgical context, methodology, and key findings.

**Table 1. Study Characteristics and Scope of Product Inventory.**

| Study | Study setting; Country | Focus of study; surgical specialty | Carbon Footprinting Approach; Guideline | Sample size | Sampling duration or year | Functional Unit: start point-end point | Included GHGs; guideline/database |
|---|---|---|---|---|---|---|---|
| [27] | 2 hospitals; Wrexham Maelor Hospital & Ysbyty Gwynedd; UK | Orthopaedics; carpal tunnel release (1) standard surgery vs (2) lean and green surgery | Process-based CF calculation; Centre of Sustainable Healthcare Guidelines | 110 carpal tunnel release surgery (7 standard surgery and 103 lean and green surgery) | 18 months (2022–2023) | 1 carpal tunnel release; | $CO_2$; Centre of Sustainable Healthcare Database |
| [17] | Stamford Hospital; USA | Obstetrics and gynaecology; Hysterectomy procedure (1) robotic assisted vs (2) laparoscopic vs (3) vaginal vs (4) abdominal | waste-focused carbon footprint analysis; nil | 100 patients (56 robotic assissted, 34 laparoscopic, 7 abdominal, 3 vaginal) | 9 months (Nov 2021-Jul 2022) | 1 hysterectomy; post-operative collection of non-reusable waste | $CO_2$; EPA emission factors |
| [14] | Orthopaedic surgery department, Institut Mutualiste Montsouris & Hôpital Pitié-Salpêtrière; Francee | Orthopaedics; arthroscopic rotator cuff repair under (1) locoregional anaesthesia (LRA) and (2) general anaesthesia (GA) | Process-based; GHG Proctocol | 52 patients (26 under LRA and 26 under GA) | 1 year Nov 2018-Apr 2019 & Nov 2020-Apr 2021 | 1 arthroscopic rotator cuff repair for a single tendon rupture; patient and staff travel to theatre, patient and staff travel home | $CO_2$; ADEME & literature emission factors |
| [21] | Large multi-center urban health system (Mass General Brigham); USA | Orthopaedics; open carpal tunnel release (oCTR) vs endoscopic carpal tunnel release (eCTR) | Process-based LCA; nil | 28 patients (14 oCTR, 14 eCTR) | 2020 | 1 carpal tunnel release procedure from OR entry to exit | $CO_2$; EPA emission factors; UK Govt landfill factors |
| [25] | Single reconstructive centre (Royal Free NHS Foundation Trust & UCL); London; UK | Plastic/Reconstructive surgery; Deep inferior epigastric perforator (DIEP) flap for autologous breast reconstruction | Process-based eco-audit (bottom-up); nil | 42 unilateral DIEP flaps (retrospective) | 17 months (Apr 2022 – Sep 2023) | 1 DIEP flap patient; first plastic surgery encounter – end of immediate postoperative recovery | $CO_2e$; GOV.UK 2023 Conversion Factors, published LCA literature |
| [75] | Welsh Centre for Burns and Plastic Surgery, Morriston Hospital; UK | Dermatology/ Plastic Surgery; Non-melanoma skin cancer excision surgery (NMSC): (1) direct closure (DC) vs (2) split-thickness skin graft (STSG) vs (3) full-thickness skin graft (FTSG) | Hybrid life-cycle assessment (process-based + EEIO); GHG Accounting Sector Guidance, PAS2050 | 32 NMSC cases (13 DC, 12 FTSG, 7 STSG) | 3 weeks (Mar 2023) | 1 NMSC; Patient arrival for day-case surgery – completion of waste collection post-surgery | $CO_2$; BEIS UK GHG Conversion factors, Greener NHS emission factors, ICE Database (v.3) |
| [12] | Large university teaching hospital; UK | Otolaryngology; adult tonsillectomy; | Process-based CF; GHG Protocol | 48 patients | 1 year (2021) | 1 adult tonsillectomy; patient and staff enter theatre-leave | $CO_2$; UK Govt conversion factors; published healthcare CF studies |

*(Continued)*

| Study | Study setting; Country | Focus of study; surgical specialty | Carbon Footprinting Approach; Guideline | Sample size | Sampling duration or year | Functional Unit: start point-end point | Included GHGs; guideline/ database |
|---|---|---|---|---|---|---|---|
| [15] | Orthopaedic University Hospital, Würzburg; Germany | Orthopaedics; 8 different procedures: total hip arthroplasty (THA), total knee arthroplasty (TKA), knee arthroscopy (KA), anterior cruciate ligament reconstruction (ACLR), shoulder arthroscopy (SA), elective foot surgery (FO), revision hip arthroplasty (RHA), revision knee arthroplasty (RKA) | Process-based bottom-up LCA; Greenhouse Gas Protocol (Scopes 1–3) | 5 procedures per operation type | Apr-Aug 2023 | 1 operation: patient enters surgical area – leaves (including acclimatized adjacent rooms) | $CO_2$, $CH_4$, $N_2O$ converted to $CO_2e$ (GWP100); local electricity/gas factors; German EF (BAFA); Anaesthetic Impact Calculator; Ecoinvent/ lit for supply chain |
| [13] | Yale-New Haven Hospital, tertiary centre; USA | Otolaryngology; standard direct laryngoscopy surgery | Process-based LCA; ISO 14040 | Not applicable (modelling study) | 2022 | 1 direct laryngoscopy surgery: | $CO_2$,$CH_4$, $N_2O$; Open LCA, ReCiPe 2016, Ecoinvent v3.7.1 |
| [76] | Ambulatory surgery center, University of Utah Health; USA | Orthopaedics; Carpal tunnel release surgery performed in (1) procedure room (PR) vs (2) operating room (OR) | Process-based LCA; US EPA, Practice Greenhealth | 14 patients (14 bilateral open carpal tunnel release with one procedure performed in each setting) | 10 years (retrospective) 2013–2023 | 1 carpal tunnel release surgery; patient enter theatre-leave | $CO_2$; US EPA, Practice Greenhealth |
| [77] | Clinique du Sport, Paris; France | Orthopaedics; Anterior cruciate ligament reconstruction (ACLR) with different techniques: (1) ST4 vs (2) BPTB (3) ITBG vs (4) HG vs (5) QTG | Process-based LCA; ISO 14040 | 10 ACLR per day per surgical crew (Modelling study) | ND | 1 anterior cruciate ligament reconstruction (ACLR); from medical prescription and decision to operate to patient discharge | $CO_2$; ReCiPe 2016, IPCC GWP100 |
| [78] | Wrexham Maelor Hospital; UK | Orthopaedics; Total hip arthroplasty (THA) (1) cemented vs (2) hybrid procedure | Process-based LCA; ISO 14040, Centre for sustainable healthcare | 20 patients (10 uncemented THA and 10 hybrid THA) | 3 months (2023) | 1 total hip arthroplasty surgery; patient enter the anaesthetic room-patient leave the recovery room | $CO_2$; Ecoinvent v3.9.1, Centre for sustainable Healthcare carbon factors |
| [79] | Yale New Haven Hospital; USA | Otolaryngology; Endoscopic sinus surgery (ESS) considering (1) landfill scenario vs (2) incineration scenario | Process-based LCA cradle-to-grave; SimaPro 9.5 | 1 ESS procedure (modelling study) | Single ESS analysed, projections for 2024 and 2030 | 1 endoscopic sinus surgery: raw material extraction, use, disposal of every material used during surgery | $CO_2$; IPCC 2021, EcoInvent, USCLI |
| [16] | Hospital for Special Surgery, New York; USA | Orthopaedics; adult spinal deformity (ASD) surgery | Process-based LCA; ISO 14040, Umberto ver. 11.9.2 | 30 patients | 5 years (retrospective data) 2017–2021 | 1 adult spinal deformity surgery; patient enter theatre-leave | $CO_2$; Ecoinvent 3.9.1 |

*(Continued)*

| Study | Study setting; Country | Focus of study; surgical specialty | Carbon Footprinting Approach; Guideline | Sample size | Sampling duration or year | Functional Unit: start point-end point | Included GHGs; guideline/database |
|---|---|---|---|---|---|---|---|
| [80] | Ambulatory surgery centre, academic hospital; USA | Orthopaedics; minor foot and ankle surgery with (1) traditional protocol vs (2) PRiSM protocol | Process-based CF calculation; nil | 40 patients (20 traditional protocol and 20 PRiSM protocol) | 5 months (2023) | One foot/ankle procedure (foreign body removal, hammertoe correction, toe amputation, hardware removal, mass excision, gastrocnemius recession); waste audit at the end of the procedure | $CO_2$; US EPA & literature emission factors |
| [32] | 4 UK hospitals (2 district general, 2 tertiary); UK | Orthopaedics; arthroscopic ACL reconstruction (ACLR) & rotator cuff repair (RCR) | Waste audit with $CO_2e$ calculation; nil | 10 patients (5 RCR and 5 ACLR for each hospital) | ND | 1 arthroscopic surgery; waste audit at the end of the procedure | $CO_2$; Waste management contractor emission factors |
| [19] | Tertiary maternity units & home births; UK and Netherlands | Obstetrics and gynaecology; (1) planned caesarean birth vs (2) uncomplicated vaginal birth in hospital vs (3) uncomplicated vaginal birth at home (comparison between UK and Netherlands) | Process-based LCA; Open LCA | ND | 2022-2023 | 1 birth of a live baby; woman enter-leave the hospital and midwives enter-leave home | $CO_2e$; Idemat 2023, Ecoinvent, DEFRA, Healthcare LCA DB |
| [30] | Single plastic surgery department, Royal Free NHS Foundation Trust; London; UK | Plastic/Reconstructive surgery; Autologous microtia reconstruction (quaternary, multistage) | Hybrid eco-audit (bottom-up for operative + top-down for inpatient stay); NHS England Sustainable Care Pathways | 23 patients undergoing autologous microtia reconstruction (5 with both stages completed, 12 only stage 1, 6 only stage 2) (retrospective) | 2 years (Apr 2022 – Apr 2024) | 1 patient pathway for multistage autologous microtia reconstruction; first plastic surgery encounter – final dressing clinic following second stage (includes 2 inpatient stays) | $CO_2e$; GOV.UK 2023 Conversion Factors, NHS England inpatient bed-day framework, published LCA literature |
| [26] | Single reconstructive centre (Royal Free NHS Foundation Trust); London; UK | Plastic/Reconstructive surgery; Immediate breast implant reconstruction | Process-based eco-audit with life-cycle analysis (bottom-up); nil | 34 immediate breast implant reconstructions (retrospective) | 13 months (Sep 2022 – Oct 2023) | 1 breast implant reconstruction patient; first appointment regarding implant surgery – end of immediate postoperative recovery | $CO_2e$; GOV.UK 2023 Conversion Factors, published LCA literature |
| [20] | Two French university hospitals – paediatric ENT departments; France | Otolaryngology; Subtotal tonsillectomy: (1) cold instruments-bipolar electrocoagulation vs (2) single-use or reusable radiofrequency electrodes vs (3) coblation | Retrospective CF calculation based on device weight, energy consumption, and production location; emission factors applied | Not stated (all subtotal tonsillectomies in 2022 using 4 techniques) | 2022 | One subtotal tonsillectomy procedure | $CO_2$; emission factors by weight, energy, origin of manufacture |
| [36] | Tertiary hospital (Amsterdam UMC, location AMC); Netherlands | Obstetrics and gynecology; Total laparoscopic hysterectomy (TLH) vs uterine artery embolization (UAE); | Process-based LCA; ISO 14040/14044; ReCiPe 2016, Hierarchist midpoint and endpoint | 40 waste inventories (10 per procedure, 8 per hospital stay, 2 per outpatient visit) | Nov 2023-Aug 2024 | 1 patient treated for uterine fibroids via TLH or UAE; outpatient to follow-up | $CO_2$; Ecoinvent v3.10; adjusted NL-specific datasets |

*(Continued)*

**Table 1.** (Continued)

| Study | Study setting; Country | Focus of study; surgical specialty | Carbon Footprinting Approach; Guideline | Sample size | Sampling duration or year | Functional Unit: start point-end point | Included GHGs; guideline/database |
|---|---|---|---|---|---|---|---|
| [33] | Regional hospital (Lismore Base Hospital); Australia | Orthopaedics; total knee arthroplasty (TKA), total hip arthroplasty (THA), ankle fracture and hand injury surgery | Waste audit + $CO_2$ emissions estimation via emission factors (0.879 $tCO_2$e/tonne waste); Australian National Greenhouse Accounts Factors | 60 procedures (15 per subtype) | Jul-Sep 2024 | 1 surgical procedure from waste generation to disposal | $CO_2$; Australian National Greenhouse Accounts Factors (2023) |
| [10] | Royal Devon University Healthcare NHS Trust; UK | Urology; Perioperative transurethral resection of bladder tumour (TURBT) pathway | Process-based cradle-to-grave LCA; ISO 14040 | 30 patients | 8 months (Dec 2022-Jul 2023) | perioperative TURBT pathway for one patient; first outpatient visit-hospital discharge | $CO_2$; Ecoinvent v3.10, Defra/BEIS |
| [18] | University Hospital of Lille (CHU Lille); France | Oral and maxillofacial surgery; Orthognathic surgery: (1) Bilateral sagittal split osteotomy (BSSO) vs (2) Le Fort I osteotomy vs (3) Bimaxillary osteotomy (Bimax) | Process-based LCA; Greenhouse Gas Protocol Scopes 1–3 | 122 patients | Jan-Jul 2024 | Patient pathway from preoperative consultation to 3 postoperative consultations; cradle-to-grave | $CO_2$; Carebone tool (AP-HP, Paris) |
| [81] | Outpatient surgical center (Bethel Park Surgical Center, UPMC); USA | Orthopaedics; Anterior cruciate ligament reconstruction (ACLR) | Process-based LCA + Material Flow Analysis (MFA) + Material Circularity Indicator (MCI); ISO 14040/14044 | 1 ACLR case (data from 3 observation sessions) | 2022-2023 | 1 ACL reconstruction, cradle-to-grave (material production to disposal/recycling) | $CO_2$; TRACI 2.1, Cumulative Energy Demand; SimaPro v9.4; Ecoinvent database |
| [11] | Radboud University Medical Centre; Netherlands | Cardiology; Coronary artery bypass grafting (CABG) | Process-based LCA cradle-to-grve; ISO 14040/14044 | 12 patients | May-June 2022 | CABG patient trajectory from OR admission to ICU discharge | $CO_2$; Ecoinvent v3.9; ReCiPe2016 method, national datasets for energy/transport |

ADEME, Agence de la Transition Écologique (France); AP-HP, Assistance Publique-Hôpitaux de Paris; BAFA, Bundesamt für Wirtschaft und Ausfuhrkontrolle (Germany); BEIS, Department for Business, Energy and Industrial Strategy (UK); CF, carbon footprint; $CH_4$, methane; $CO_2$, carbon dioxide; DEFRA, Department for Environment, Food & Rural Affairs (UK); EEIO, environmentally extended input–output; EF, emission factor(s); EPA, Environmental Protection Agency; GHG, greenhouse gas; GWP100, global warming potential over 100 years; ICE, Inventory of Carbon and Energy database; IPCC, Intergovernmental Panel on Climate Change; ISO, International Standards Organization; LCA, life cycle assessment; NHS, National Health Service (UK); PAS 2050, publicly available specification for assessing life cycle greenhouse gas emissions; TRACI, Tool for the Reduction and Assessment of Chemical and other Environmental Impacts; UMC, University Medical Center; UPMC, University of Pittsburgh Medical Center.

Geographically, the studies were conducted in the United Kingdom (10 studies), United States (8 studies), France (4 studies), Netherlands (2 studies), Germany (1 study), and Australia (1 study), with some collaboration across countries in a few cases. This spread reflects a concentration of interest in countries with active healthcare sustainability initiatives (e.g., NHS England's net-zero plan, academic centers in the US like Yale and University of California focusing on medical LCAs, etc.) and in Europe (often motivated by hospital or national carbon reduction goals).

There is an uneven distribution of published studies across surgical specialties, with orthopaedic surgery being the most frequently investigated (12 studies), whereas most other specialties are represented by only one or a few studies.

The carbon footprint range results reported in each study are summarized in Fig 2. Overall, the reported $CO_2$e emissions associated with surgical procedures span a wide range across the literature, from values close to 1 kg$CO_2$e (for

**Table 2. Comparison of Inventory Boundaries.**

| Process | | [27] | [17] | [14] | [21] | [25] | [75] | [12] | [15] | [13] | [76] | [77] | [78] | [79] | [16] | [80] | [32] | [19] | [30] | [26] | [20] | [36] | [33] | [10] | [18] | [81] | [11] |
|---|---|---|---|---|---|---|---|---|---|---|---|---|---|---|---|---|---|---|---|---|---|---|---|---|---|---|---|
| Travel | Staff | | | × | × | × | | × | | | | | | | | | | | × | × | × | × | | × | × | | × |
| | Patient | | | × | × | × | | × | | | | | | | | | | | × | × | × | × | | × | × | × | × |
| Energy & Natural Resources | Electronic equipment | | | × | × | × | × | × | × | | | × | × | × | × | | | × | × | × | × | × | | × | × | × | × |
| | HVAC | | | × | × | × | × | × | × | | × | × | × | | × | | | × | × | × | × | × | | × | × | × | × |
| | Lighting | | | × | × | × | × | × | × | | × | × | × | | × | | | × | × | × | × | × | | × | × | × | × |
| | Water | | | | | × | | | | | | × | × | × | | | | | × | × | × | × | | × | × | | × |
| Consumables | Manufacturing and Procurement | × | | × | | × | × | × | × | × | × | × | × | × | | × | | × | × | × | × | × | | × | × | × | × |
| | Transport | | | | | × | | | | × | | × | × | × | × | | | × | × | × | × | × | | × | | × | × |
| Medical Devices | Manufacturing and Procurement | × | | × | | × | × | × | × | × | | | × | × | × | | | × | × | × | × | × | | × | × | × | × |
| | Transport | | | | | | | | | | | × | × | × | × | | | | × | × | × | × | | × | | | × |
| Reusables | Manufacturing and Procurement | × | | × | | × | × | × | × | × | | × | × | × | | | | × | × | × | × | × | | × | × | × | × |
| | Transport | | | | | × | | | | × | | × | × | × | | | | | × | × | × | × | | × | | | × |
| Reprocessing | Sterilization | × | | × | × | × | × | × | × | × | × | × | × | × | × | × | | × | × | × | × | × | | × | × | × | × |
| | Laundry | × | | × | × | × | × | × | × | ? | × | × | × | ? | | × | | × | × | | × | × | | × | × | × | × |
| Pharmaceuticals | Manufacturing and Procurement | | | × | | | × | × | | | | | | × | | | | × | | | × | × | | | × | | × |
| | Transport | | | | | | | | | | | | | × | | | | | | | | × | | | | | |
| | Anaesthesia | | | × | | × | | × | × | × | | × | | × | × | | | × | × | × | × | × | | × | × | | |
| Waste management | Transport | | | × | | × | × | × | × | | | × | × | × | | × | | | × | | × | × | × | × | × | × | × |
| | Disposal | × | × | × | × | × | × | × | × | × | × | × | × | × | × | × | × | × | × | × | × | × | × | × | × | × | × |

An 'x' indicates that the process was explicitly included within the study boundary, while blank cells denote omission; question marks ('?') indicate uncertainty or unclear reporting

certain low-resource tonsillectomy techniques and other minimal scenarios) up to 414 kgCO$_2$e per case for a CABG procedure in a Dutch hospital. When entire multistage care pathways are considered, the upper bound rises substantially: the eco-audit of autologous microtia reconstruction, a quaternary plastic-surgery proceure spanning two operative stages, multiple pre-operative and follow-up visits, and two inpatient admissions, reported approximately 1,005 kgCO2e per patient. Most routine elective surgeries (with general anaesthesia in a hospital setting) tended to fall in the range between a few kilograms to a few hundred kilograms of kgCO$_2$e.

The typical carbon footprint of a common elective surgery (e.g., knee arthroplasty, laparoscopic hysterectomy, etc.) appears to be on the order of 50–150 kgCO$_2$e, whereas more minor procedures or those done under streamlined protocols can be less than 30 kgCO$_2$e, and very complex or resource-intense procedures can be more than 400 kgCO$_2$e.

The system boundary for each included study was mapped to identify which processes were accounted for in the carbon footprint assessment. Table 2 summarizes the inclusion or omission of key process categories, such as travel, energy use, consumables, medical devices, reprocessing, pharmaceuticals, and waste management, across studies. When specific processes were excluded, the resulting footprint likely underestimates the total environmental impact of the surgical pathway. This comparison highlights substantial heterogeneity among studies in terms of which stages were considered, suggesting that results are not directly comparable when inventory boundaries differ.

## Emission hotspots in surgical operations

Despite the diversity of procedures and methods, the studies overwhelmingly point to a few common GHG emission hotspots in the surgical process.

**Medical devices, surgical supplies, and implants.** The production and use of single-use disposable items and other consumables emerge as the top contributor to surgical carbon footprints in 17 of 26 studies. For example, consumables accounted for 22% of the TURBT pathway emissions in the UK [10] and 39% of total emissions in CABG procedures [11]. Similarly, in the tonsillectomy LCA, consumables contributed 17%, the largest single category after utilities [12]. This group encompasses surgical gloves, gowns, drapes, single-use instruments or device components, procedure packs, synthetic implants (mesh, screws), and certain pharmaceuticals, with emissions arising from the entire upstream supply chain (raw material extraction, manufacturing, sterilization, packaging) and the downstream disposal (often incineration). Comparative analyses further highlight their impact: in direct laryngoscopy, reusable instrument kits contributed less than 1 kgCO$_2$e, while sterile single-use supplies added approximately 7 kg CO$_2$e per procedure, corresponding to nearly 90% of the total footprint [13]; in shoulder arthroscopy, single-use anchors and suture devices were identified as principal drivers of emissions, and their partial replacement achieved roughly a 12% reduction in total carbon output [14]. A German assessment of eight orthopaedic procedures similarly identified consumables as the largest source, accounting for 34.6% of emissions [15]. In spine fusion, metallic implants and bone graft materials were among the most carbon-intensive components due to manufacturing and material processing [16]. Even when comparing abdominal hysterectomy with uterine artery embolization, the higher footprint of the surgical route was partially attributable to more resource-intensive surgical supplies (staplers, sutures, and mesh) compared with interventional materials (coils, catheters) [17].

**Energy use (HVAC and lighting).** Hospitals run energy-intensive HVAC systems to maintain strict air exchanges and climate control in operating rooms. Many studies pointed to electricity and natural gas use for OR ventilation, heating, and lighting as a major hotspot. One analysis of tonsillectomy procedures found utilities (mainly HVAC) to be the largest source of GHGs (over 50%) [12]. In another CABG LCA, energy use contributed 48 kgCO2e (12%), and they note most of that was from OR HVAC systems [11]. In maxillofacial surgery, energy use was a smaller contributor (0,5%−0,8%) compared with patient travel, which dominated the total footprint [18]. The contribution of energy can vary substantially depending on the type of energy carrier used (e.g., electricity, natural gas, district heating) and, in the case of electricity, on the carbon emission factor of the national grid. For example, heating with natural gas has broadly similar carbon intensity across countries, whereas electricity-related emissions differ widely: a kWh in France, largely produced from

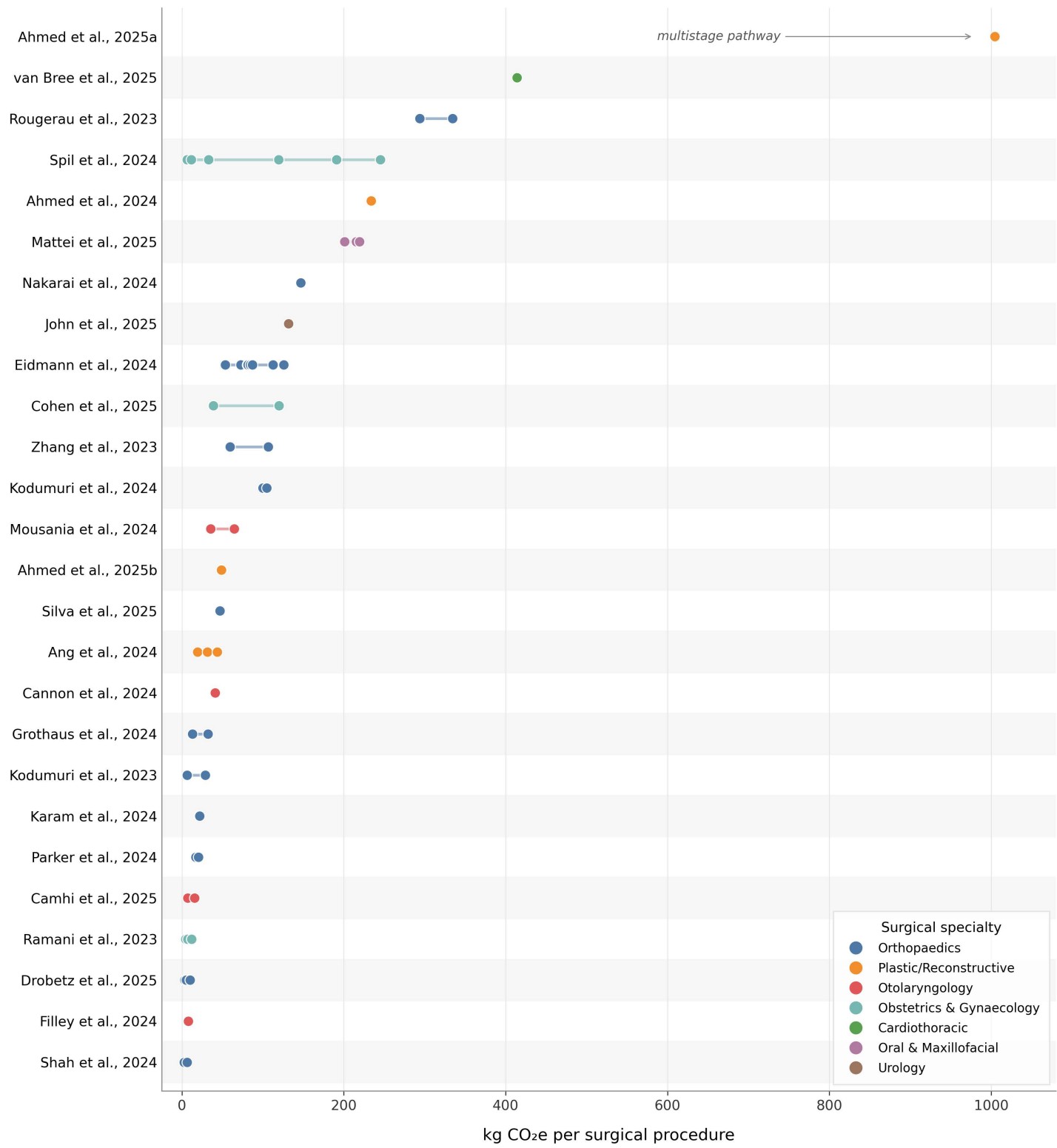

**Fig.2. Reported carbon footprints of surgical procedures across the 26 studies included in the systematic review (2023-2025), expressed in kgCO$_2$e per surgical procedure.** Each row corresponds to one study; markers indicate point estimates and connecting lines indicate the range when

multiple values were reported, reflecting comparative arms (e.g., anaesthetic technique, surgical approach) or different procedure variants within the same study. Markers are colour-coded by surgical specialty. Studies are ordered top-to-bottom by their maximum reported value.

nuclear energy, has a much lower emission factor than in countries with coal or gas-intensive grids. This might explain why, for example, the French tonsil study doesn't mention energy as a factor (likely low impact per case, and they focused on devices) whereas the UK and NL studies do [19–21]. Nonetheless, keeping an OR running (lights, anaesthesia machine, monitors) for a procedure has a non-trivial footprint, and when OR time or efficiency is studied, it connects to this. For instance, endoscopic carpal tunnel release surgery (CTR) had a higher footprint than the open approach largely because of its longer OR time (hence more facility energy use) than open CTR [21]. Facilities-related emissions in that study were a big portion of the 83 kgCO$_2$e (meaning HVAC, lights, etc.).

**Anaesthetic gases and pharmaceuticals.** The use of volatile anaesthetic agents (such as desflurane, sevoflurane) and nitrous oxide (N$_2$O) can be a major GHG contributor because these gases are themselves potent greenhouse gases when released unscavenged [22–24]. However, their impact in the carbon footprint results varies widely depending on if and how studies included them. In several recent studies, anaesthetic gases were purposefully minimized or omitted: one study demonstrated that eliminating volatile gases via regional anaesthesia reduced the footprint by 40.9 kgCO$_2$e (which we infer largely came from eliminating desflurane) [14]. Another assessment of labour analgesia illustrated the extreme case with nitrous oxide: just that gas usage can dwarf all other sources, adding 230 kgCO$_2$e per case [19]. In contexts like standard surgery under general anaesthesia, the footprint of anaesthetic gases depends on which agents are used and for how long. Desflurane has a GWP c.a. 2,500 times CO$_2$ and N$_2$O c.a. 300, so even a few litres can equal tens of kg of CO$_2$. Anaesthesia/drugs and associated adjuncts were 27% of TURBT pathway emissions, although they clarify much of that was from consumables in anaesthesia (e.g., disposable breathing circuits, meds) rather than the gases themselves [10]. Indeed, a trend is that many centres have shifted away from high-emission gases: the CABG study mentioned total intravenous anaesthesia (TIVA) was used, so anaesthetic gases didn't feature heavily in that footprint [11]. In contrast, where older practices involved desflurane or N$_2$O, these agents typically dominated the total emissions profile. Overall, anaesthetic practice remains a controllable hotspot: when high-GWP gases are employed, they can be the largest single contributor to surgical emissions, whereas adoption of low-GWP or gas-free approaches markedly reduces this impact, though drug production and supply chains still contribute residual emissions. A particularly striking example comes from plastic and reconstructive surgery: in DIEP flap breast reconstruction, the induction, maintenance and running of anaesthesia together contributed approximately 67.6% of the total procedural footprint (158 of 234 kgCO$_2$e), reflecting the long operative time (typically 8–10 hours) characteristic of microvascular procedures and the use of nitrous oxide plus oxygen at 5 L/min for maintenance [25]. In contrast, the breast implant reconstruction pathway at the same centre [26] adopted total intravenous anaesthesia (propofol + remifentanil) with no volatile maintenance gases: under this protocol, anaesthesia-related emissions fell to 16.6% of total procedural footprint, illustrating in practice how the choice of anaesthetic technique alone can reduce this hotspot by several-fold within the same clinical setting.

**Patient and staff travel.** Travel emissions can be significant, especially in studies with broader pathways. In maxillofacial surgeries, patient transportation accounted for 66–70% of total emissions, because patients travelled for multiple consults and follow-ups for their surgery [18]. In an assessment of coronary artery bypass grafting, staff commuting contributed around 9% (36 kg CO$_2$e) of total emissions, ranking as the third-largest source after consumables and energy use [11]. Patient travel for pre-op was 18.6% of TURBT emissions [10]. In comparative assessments of childbirth in different European countries, travel mode and distance contributed to inter-country variability, although their influence was overshadowed by the dominant impact of nitrous oxide use [19]. Travel didn't feature in some studies that only considered OR time [21,27], but where it is counted, travel often emerges as a major emissions source, especially when the procedure itself has relatively low inherent emissions or when multiple visits are required. This underscores

the importance of defining system boundaries clearly: decarbonizing hospital operations alone may not substantially reduce overall emissions if extensive patient or staff travel persists. The integration of telemedicine, consolidation of appointments, and scheduling strategies to minimize round trips have been proposed as practical interventions [28,29]. It is context-dependent: a single outpatient visit might be small, but numerous visits for complex care can add up to more than the surgery itself [18]. Similar findings were reported for autologous breast reconstruction: in the DIEP flap pathway, combined patient and staff travel contributed more than 15% of total emissions [25], and in the autologous microtia reconstruction pathway patient travel alone represented the dominant source when excluding inpatient stay (83.4% of bottom-up calculated emissions), reflecting the geographical centralization of quaternary plastic surgery services [30].

**Sterilization and reprocessing.** The energy and water needed to sterilize reusable instruments or linens contributes some emissions. In tonsil surgery, instrument decontamination was 5.4% (2 $kgCO_2e$ of the 41 $kgCO_2e$) [12]. One study specifically tested an alternative to steam sterilization (high-level disinfection) and found it could cut 11% of the procedure's GHG by saving energy [13]. Although sterilization is not typically the dominant contributor, it remains a consistent component of surgical footprints. Transitions from reusable to single-use systems essentially shift emissions from sterilization to manufacturing processes, often increasing overall impacts, though results depend on usage rates and local energy sources. Generally, life cycle analyses favour reusable instruments when reprocessed a sufficient number of times, since the manufacturing of a new disposable item for each procedure tends to be far more carbon-intensive than repeated steam sterilization cycles

It should be noted that sterilization and laundry emissions are frequently approximated in surgical LCAs rather than directly measured. Many studies rely on standardized emission factors from reference assessments [31], which provided baseline energy and water consumption estimates per reprocessing cycle for orthopaedic instruments. This practice, while practical for comparability, introduces some uncertainty that should be considered when interpreting absolute values.

**Waste disposal.** Paradoxically, disposal of waste (incineration or landfill) was addressed in each article reviewed, but its impact is usually a minor contributor to GHG. In several analyses, waste treatment accounted only 3–5% [11,12]. Evidence from operating room audits confirms that focusing exclusively on waste treatment captures less than 10% of the total footprint [32,33]. Annual waste-related emissions for surgical departments have been measured in the range of a few tonnes of $CO_2e$, whereas upstream emissions from procurement, manufacturing, and energy use reach tens to hundreds of kilograms per individual procedure. Nevertheless, waste management remains environmentally relevant beyond its carbon impact due to plastic pollution, resource depletion, and the visibility of waste streams within healthcare facilities. Moreover, regulated medical waste incineration carries a substantially higher $CO_2$ emission factor per kilogram than general waste. Multiple analyses emphasized that improved waste segregation could prevent up to 90% of non-hazardous materials from entering high-energy treatment streams unnecessarily. The hotspot is not the incineration emissions per se, but the choice of disposable vs reusable which affects how much waste is generated in the first place.

The major carbon hotspots identified across studies are: (1) the supply chain of single-use devices and consumables, (2) energy use for OR HVAC/equipment, (3) travel (in broader system-boundary analyses), (4) anaesthetic gases (where used), and (5) to a lesser extent, sterilization processes and waste management. These broadly correspond to what the 2022 Green Theatre Checklist initiatives have targeted and confirmed in 2025: focusing on anaesthetic practices, energy, procurement, and waste segregation. A qualitative systematic review also found anaesthetic gases, sterilization, and habitual practices (like leaving equipment running) as key emission sources, and linked many sustainability measures to structural changes like policy on reusables and recycling [34].

## Mitigation strategies and sustainability initiatives

A central goal of these studies, and of this review, is to identify how we can mitigate the carbon emissions of surgical care. The included studies not only quantified footprints but often discussed or even tested interventions to reduce them. The summary of mitigation strategies shown in Table 3 highlights how research into surgical sustainability is gradually evolving

**Table 3. Mitigation Strategies Proposed Across Studies.**

| Category | Mitigation strategy | Description and Evidence | Estimated/ Reported Impact |
|---|---|---|---|
| Anaesthetic practice optimization | Avoid high-GWP gases (N$_2$O, desflurane) [14] | Elimination of high-GWP agents markedly reduces emissions; N$_2$O use dominates birth-related procedures. | N$_2$O adds 230 kg CO$_2$e per delivery; avoidance reduces > 90% of anaesthesia-related emissions. |
| | Prefer regional or local anaesthesia [14] | Regional blocks eliminate volatiles and infrastructure demand; maintains outcomes. | 12% lower footprint in arthroscopic rotator cuff repair vs general anaesthesia. |
| | Optimize fresh-gas flows (low-flow) [12] | Low-flow techniques cut volatile agent use by > 70%; recommended in guidelines. | Qualitative reduction in agent consumption. |
| Rationalization of surgical supplies | "Lean" instrument trays and custom packs [27] | Removing unused items and minimizing kit contents drastically cuts waste and emissions. | −65% waste; −80% CO$_2$e per CTR case. |
| | Prefer reusable hardware and linens [12,32,33,81] | Replacing disposables with reusable metal tools, drapes, and gowns reduces supply-chain emissions. | 30-50 kgCO$_2$e saved per case (depending on procedure). |
| | Optimize material and implant choice [77,81] | Selection of biocompatible, low-energy-manufacture materials for implants and grafts. | 10-15% reduction in procedure footprint. |
| | Limit oversupply/ "open but not used" [10,27] | Customize packs; open only needed items. | 5-15% reduction in supply-chain emissions. |
| Reusables and reprocessing | Reuse of anaesthesia circuits and devices [10] | Substituting disposable with reusable circuits reduces plastic waste. | Qualitative reduction documented. |
| | Third-party reprocessing of single-use devices [75,79] | Certified reprocessing of high-value devices (e.g., harmonic scalpels). | Up to 50% reduction vs new device. |
| | Linen reuse optimization [13,32,81] | Smaller drapes/towels; cloth instead of plastic when safe. | 28% water saving; less waste volume. |
| Energy and facilities | Optimize OR HVAC and equipment use [10,15,81] | HVAC is a major hotspot; reduce air exchange when idle and power-down equipment. | 10-20% energy savings; proportional CO$_2$ reduction. |
| | Transition to renewable energy sources [11,81] | Modelling shows that green electricity significantly lowers hospital footprints. | −13% to −17% overall in modelled scenarios. |
| Travel and logistics | Reduce staff and patient travel; telehealth [10,18] | Scheduling optimization and virtual consults cut transport emissions. | Patient travel 18–70% of pathway in broad LCAs; savings proportional to visits avoided. |
| Sterilization and decontamination | Optimize reprocessing cycles and methods [12,15] | High-level disinfection instead of steam can reduce energy demand; review sterilization load per reusable use. | ENT tonsillectomy: 5% of total CO$_2$e; alternative method cut 11%. |
| Waste management | Segregation and recycling programs [32,33] | Proper sorting prevents clean plastic from incineration; improves recycling rates. | 44% of OR waste potentially recyclable; high CO$_2$ saving per kg diverted. |
| | Source reduction > disposal focus [10,12] | Majority of emissions stem from production, not incineration. | > 90% of footprint upstream (materials + energy). |
| Surgical process optimization | Improve OR list utilization and day-case management [10,20] | Efficient case scheduling reduces per-case overhead; outpatient pathways lower ward energy load. | 10-20% per case CO$_2$e reduction with same day discharge vs inpatient. |
| Education and cultural change | Staff engagement and training [32,33,81] | Education on waste segregation, lean practice and sustainable choices is key for implementation. | Non quantified but recognized as essential enabler for behavioural change. |

from a purely descriptive approach (quantification of carbon footprint) towards an interventionist approach, geared towards identifying concrete actions. Most of the proposed strategies are behavioural and organisational rather than technological: reducing anaesthetic gases with high global warming potential, rationalising surgical sets and increasing the use of reusable materials are measures that can be implemented within the operating theatre without requiring complex structural or organisational changes.

Systemic measures, such as the energy transition to renewable sources or the reorganisation of the waste cycle, are mostly modelled or discussed in theoretical terms, indicating the need for greater institutional commitment. The growing integration of environmental sustainability into clinical decision-making reflects a cultural shift towards considering environmental impact as an intrinsic dimension of quality of care. The evidence reported in the studies supports a hierarchy of actions for the decarbonisation of surgery in which behavioural and organisational optimisation precedes infrastructural transformation, offering a bottom-up approach.

**Quality assessment and risk of bias**

The quality scores for each individual study were calculated following the approach adopted in previous studies [6,7], and the results are reported in S1 Table. Four main domains were assessed: completeness, consistency, transparency, and accuracy each comprising specific sub-criteria. These quality domains align with core principles of greenhouse gas accounting which are emphasized in established carbon footprinting standards (e.g., ISO 14040/14044 and the GHG Protocol). For each item, studies were scored on a three-point ordinal scale (2, 1 or 0) according to predefined criteria. Completeness considered the extent to which system boundaries and GHG scopes were fully captured. Consistency evaluated adherence to recognized carbon footprinting guidelines and uniformity of methodological application in comparative analyses. Transparency assessed the clarity of stated hypotheses and objectives, the reporting of included GHGs, assumptions, exclusions, data completeness, and the level of numerical detail in reported results. Accuracy examined the specificity of data sources, and the consideration of both parameter and scenario uncertainty, including whether confidence intervals (CIs) were reported. Each study's total score (maximum = 24, or 22 when items were not applicable) was converted to a percentage to facilitate comparison of relative quality across studies. Higher percentages indicate greater methodological rigor and reporting transparency.

The methodological quality of the 26 included studies varied substantially, with overall quality scores ranging from roughly 38% to 100% of the total criteria. In general, more recently published studies (2024–2025) tended to score higher, reflecting a trend toward improved methodological rigor and adherence to reporting guidelines.

**Variability in system boundaries.** A major source of heterogeneity was the definition of system boundaries which varied along two distinct dimensions: the breadth of GHG scopes considered, and the temporal extent of the patient pathway included. Considering first the GHG scope dimension, only about half of the studies explicitly accounted for all three scopes of healthcare GHG emissions (see S2Table for details). In contrast, many studies employed truncated boundaries, omitting one or more scopes. These boundary differences pose a significant comparability challenge, a "carbon footprint" of a procedure in one study might encompass a very different set of processes than in another study. As a result, absolute footprint values cannot be directly compared across studies without accounting for what sources were included. Notably, the highest footprint estimates tended to come from studies with the broadest system boundaries, whereas narrowly focused studies reported much smaller carbon totals. This underscores that boundary scope is a key determinant of the reported footprint. It also represents a potential bias: studies that neglect major upstream or downstream processes are likely to underestimate total emissions, sometimes substantially so. Given that 70–80% of healthcare's carbon emissions typically arise indirectly along the supply chain (Scope 3) [35], excluding these components can omit the majority of a surgery's true footprint. Encouragingly, several of the most recent studies did adopt more comprehensive life-cycle boundaries that include supply-chain contributions, aligning with best-practice recommendations to broaden scopes in healthcare sustainability metrics. Variability in system boundaries remains a prominent limitation of the evidence base, and greater standardization is needed to ensure completeness and enable fair comparisons.

A second, equally consequential boundary dimension concerns whether the carbon footprint is estimated for the perioperative phase only (typically from entering to leaving the operating theatre) or for the whole patient care pathway, encompassing preoperative consultations, imaging, travel, inpatient stay and postoperative follow-up. This distinction has profound implications for both the magnitude and the composition of the reported footprint. Narrow-boundary studies

focused on operating-theatre activity [21,27] tend to report lower totals dominated by consumables and intraoperative energy use, while whole-pathway assessments [10,11,18,25,26,30,36] capture additional sources such as patient and staff travel, outpatient clinic energy use and inpatient bed-days, which can individually exceed the emissions of the surgical step itself. Recognizing this dichotomy is essential when interpreting comparative figures: whole-pathway assessments are more representative of the true climate burden of surgery, while perioperative-only audits are more suitable for evaluating intraoperative interventions.

**Transparency of assumptions and reporting.** Most studies were relatively strong in reporting transparency, but there were notable exceptions. Nearly all clearly stated their overarching goals and scope (e.g., defining the surgical procedure and system studied), and 90% explicitly listed which greenhouse gases were counted (typically $CO_2$, with some also including $CH_4$ and $N_2O$ in $CO_2$e) in their calculations. Many also provided detailed breakdowns of results by subprocess or emission source, which enhances interpretability. This level of detail is a strength, as it allows readers to see the major contributors and check consistency with the methods. On the other hand, a minority of studies lacked clarity in important assumptions. In roughly one-third of the papers, the authors did not clearly document the number of data points or measurements underlying each process, for instance, whether energy use was measured directly or estimated, and how many observations informed that estimate. Such omissions make it difficult to judge the reliability of those results. In a few cases, key methodological choices were only vaguely described or buried in supplementary materials (e.g., how equipment utilization was allocated, or why certain items were excluded). The exclusions and assumptions reported by each study (see S3 Table) reveal that while most authors did state their major assumptions and justifications, some provided only limited detail. None of the included studies underwent any external peer audit of their carbon accounting, and only a handful explicitly discussed the completeness of their data or potential gaps. This lack of critical appraisal of data completeness is common in healthcare LCAs, previous work has found that authors often do not assess whether all relevant processes/data were captured [37]. Overall, transparency was one of the stronger areas of quality (reflected in consistently high scoring on this domain), but there remains room for improvement in clearly conveying all assumptions, exclusions, and data collection methods. Clearer reporting would improve reproducibility and allow the research community to more easily identify reasons for differences between studies.

**Data sources and quality.** The accuracy and reliability of the footprint estimates depend heavily on data quality. In this regard, the studies showed mixed performance. Only 2 out of 26 studies relied exclusively on primary data (i.e., direct measurements specific to the study hospital) for all inputs considered; the vast majority combined primary and secondary data. Commonly, authors measured a few parameters in situ and obtained others from the literature or databases. While using secondary data is often unavoidable, for example, no single hospital can directly measure the embodied emissions of manufactured drugs or devices, it does introduce uncertainty and potential bias if the data are not representative. Many studies assumed literature values or industry averages that may not reflect the local context of their hospital. For instance, several analyses used national conversion factors for waste disposal or energy, sometimes even from other countries due to data availability. Few studies discussed the implications of these choices. In fact, consistency in data representativeness was generally weak: none of the studies formally evaluated how temporally or geographically representative their data were.

Many studies had to assume usage patterns (for equipment, HVAC, etc.) in the absence of direct monitoring. The specificity of data sources was captured in our quality scoring, nearly all studies scored "1" (on a 0–2 scale) for using a mix of primary and secondary data, indicating moderate fidelity. Importantly, the lack of uncertainty analysis means we do not know how much error these data choices might be introducing. In summary, data quality is a concern across the board. Even where primary data were used, sample sizes were often small: some studies based calculations on just 5–15 procedures, which raises questions about statistical reliability. Secondary data, while necessary for life-cycle modelling, were not always transparently sourced or up-to-date. These issues highlight the need for more robust data collection and validation in future studies, as well as for reporting of data uncertainties.

 

**Adherence to LCA standards.** About half of the studies explicitly stated that they followed a recognized carbon footprinting or LCA standard, whereas the others did not mention any formal guidance. This split was reflected in the consistency domain scores. Those that did cite a standard (such as the ISO 14040/14044 LCA framework or the GHG Protocol Product Standard) generally showed better methodological coherence. For example, several studies referenced using the GHG Protocol or ISO guidelines to define their scope and inventory, and one study applied a < 1% materiality threshold (cut-off rule) for minor emission sources, which is in line with PAS 2050 and other standards. By contrast, studies that did not reference any guidelines sometimes made ad hoc methodological choices, for instance, defining functional units or impact boundaries in non-standard ways, which could introduce bias. In our sample, there was evidence that newer studies are moving toward accepted norms. Many 2024–2025 studies referenced, at minimum, the inclusion of Scope 1–3 emissions per the GHG Protocol taxonomy or cited performing a life-cycle assessment per ISO protocols. This is a positive development because following established frameworks tends to improve consistency: it ensures, for example, that if two studies both claim to do an LCA, they will at least share common definitions and steps (goal and scope definition, inventory analysis, impact assessment, interpretation). Indeed, among the comparative studies in our review, nearly all applied uniform methods to both comparators, yielding high consistency scores. Only one or two comparative studies had minor consistency issues (e.g., slightly different data sources used between the compared groups), and these were flagged as potential biases. Overall, the use of standard methods is a notable strength for roughly half the studies; the remainder would benefit from closer alignment with published LCA guidance to enhance credibility.

## Discussion

Compared with the two prior systematic reviews on this topic, which included 8 and 7 studies respectively [6,7], the present update incorporates 26 eligible works published in just three years (2023–2025), reflecting the rapid growth of the field. Reported footprints now span from under 5 kgCO$_2$e for minor procedures to over 400 kgCO$_2$e for complex single-stage operations, with whole multistage patient pathways reaching about 1,000 kgCO$_2$e. Beyond the larger evidence base, the current corpus differs from the prior reviews in three substantive ways: it covers a markedly broader set of specialties (including cardiothoracic, urologic, obstetric, maxillofacial and plastic/reconstructive surgery), a more diverse geographical distribution, and a larger proportion of studies adopting formal LCA frameworks (ISO 14040/14044, GHG Protocol).

Across procedures and specialties, single-use instruments and consumables consistently emerge as the dominant hotspot, confirming previous evidence that material production and sterilization for disposables represent a substantial proportion of each procedure's footprint. Several recent studies compare the environmental impacts of single-use versus reusable surgical and medical instruments [38–44]. In many cases reusable instruments do offer lower overall emissions, but there are situations where the advantage is marginal or context-dependent, reminding us that "reusable" is not automatically synonymous with "sustainable" without proper implementation. Operating rooms are energy-intensive environments due to ventilation, air conditioning, heating, and specialized equipment, so electricity use and HVAC requirements often make up the next largest share of emissions after materials, underscoring the need for both local efficiency measures and institutional decarbonization through renewable energy sources. This is a central issue not only for operating rooms but for hospital buildings in general: there is a growing concern in other high-intensity departments like diagnostic imaging suites (e.g., MRI and CT scanners) and clinical laboratories as well [45–49].

When included, travel emissions from patients and healthcare staff add a significant layer of impact, highlighting that system-level organization and telehealth integration can reduce emissions beyond the operating room itself. Better coordination of care to minimize unnecessary trips, and the integration of telehealth services, can avoid a great deal of travel-related emissions [28,29,50–52]. Anaesthetic practices also influence results markedly: eliminating desflurane and nitrous oxide, adopting regional techniques, and using low-flow anaesthesia can achieve immediate reductions without

compromising safety. The literature in recent years has been rich with papers documenting the outsized climate footprint of anaesthetic gases and highlighting successful interventions, from hospital case studies eliminating desflurane to national policies (the EU's decision to ban desflurane as of 2026) [22–24,53–57].

Reprocessing and waste management, although smaller in relative magnitude, remain critical in the transition from disposable to reusable systems. Effective waste segregation, recycling programs, and reprocessing of medical devices can all help decrease the environmental toll of surgical care and foster a more circular use of materials. In recent literature, an increasing number of studies focus on waste reduction strategies and circular economy principles in surgical contexts, for example, optimizing surgical kits to reduce unused items, recycling polypropylene plastics from instrument trays, or reusing sterilized equipment where feasible [58–63]. These efforts not only cut emissions marginally but also have broader environmental benefits (like reducing landfill waste and pollution) and often yield cost savings for hospitals.

A persistent challenge highlighted by both the present synthesis and the prior reviews is the marked methodological heterogeneity that limits cross-study comparability. Convergence on a small set of shared principles would substantially accelerate cumulative learning in this field. Future studies should explicitly adopt a recognized LCA framework, such as ISO 14040/14044 or the GHG Protocol Product Standard, and report transparently their system boundaries, functional units and data sources, declaring which life-cycle stages are included and which are excluded. Systematic uncertainty analysis should accompany the headline figure, and whole-pathway audits should be preferred over perioperative-only assessments whenever the clinical context allows. Reporting checklists tailored to healthcare LCAs would further support consistency across studies. The three plastic surgery eco-audits [25,26,30] illustrate this replicable approach in practice: by applying the same process-mapping methodology, emission-factor sources and reporting structure to three very different procedures (DIEP flap, breast implant and multistage microtia reconstruction) at a single centre, the team produced internally comparable results that make visible how boundary choices, operative duration and anaesthetic technique drive the total footprint, offering a template that could be extended to other specialties and institutions. Building on these methodological considerations, the CABG study [11] and the minimally invasive gynaecological pathway assessment [36] serve as useful reference points for future work, since both combined process-based bottom-up data collection with broad system boundaries, covered multiple emission scopes including supply chain and travel, applied standardized emission factors and ISO-compliant LCA steps, and explicitly discussed parameter uncertainty.

Beyond the technical agenda, the evidence reviewed reinforces that achieving low-carbon surgery requires coordinated behavioural, organizational and infrastructural strategies, with environmental performance treated as an intrinsic dimension of surgical quality rather than an isolated objective. It's also worth noting that environmental sustainability is becoming an important competitive and reputational concern in the healthcare industry. Hospitals that adopt sustainable methods are increasingly seen favourably by environmentally concerned patients, employees, and investors. In an era of increasing public awareness of climate change, many healthcare providers understand that displaying climate leadership can boost community trust and appeal [3,64–66]. Some healthcare systems increasingly publish sustainability scores or get awards for green activities, and procurement processes are beginning to favour producers of low-carbon items. This means that sustainability is more than just an ethical necessity or a regulatory compliance issue; it is also regarded as a symbol of quality and innovation in healthcare. Real advancement, however, is dependent on culture and mindsets as much as technology and policy [67,68]. Fortunately, various surveys have been undertaken to assess the impression of environmental sustainability among healthcare professionals, and they generally demonstrate a great hunger for change on the ground. When clinicians and hospital workers are taught about the environmental impact of healthcare operations, they strongly support initiatives to decrease waste and emissions [24,69–71]. Many front-line employees are ready to contribute to greener practices if given the opportunity and assistance. This bottom-up engagement is critical: while high-level promises and management initiatives lay the groundwork (the "top-down" part), long-term change frequently occurs only when the workforce is fully aware, engaged, and empowered to act [72–74]. Building environmental awareness into the culture of surgical teams and hospitals, through

education, training, and recognition of sustainability achievements, can thereby generate substantial grassroots improvements that eventually transform high climate goals into daily practice. The climate issue in healthcare will be addressed not simply by administrative decisions or new technologies, but also through the collective activities of many people rethinking ordinary practices and challenging what is currently in place.

Despite the advances in knowledge and practice outlined above, there remain notable gaps in the literature that future research needs to address. High-volume or common types of surgery are still under-represented in detailed carbon assessment (e.g., appendectomy, cholecystectomy, neurosurgery, and paediatric surgery). The environmental impact of emerging technologies like robotic surgery has only been partially explored: while waste data and other proxies suggest that robotic procedures have higher footprints, detailed LCAs comparing robotic and conventional surgery are still needed. Similarly, the footprint of implant production (orthopaedic hardware, surgical mesh, etc.) is an area needing deeper analysis; some of our included studies touched on it qualitatively (e.g., spinal implants), but precise data would help target greening efforts in the medical device industry (e.g., using low-carbon materials or recycling metals).

Another knowledge gap is long-term outcome and carbon trade-offs: for instance, if a minimally invasive approach has a higher surgical footprint but leads to fewer complications or re-operations, the net emissions over a patient's episode of illness might favour one or the other. A life-cycle perspective on a patient's treatment course (including potential downstream events) would align carbon accounting with clinical outcome accounting. None of the studies did a full life-cycle cost analysis combining environmental and clinical outcomes (e.g., emissions per quality-adjusted life year, QALY), but that could be a valuable approach for future research, potentially informing policy.

It is worth noting that while individual changes can make significant relative differences, tackling climate change requires systemic action. The cumulative impact of widespread adoption of these changes could be substantial: the healthcare sector is responsible for 4–5% of global GHG emissions, and within that, operating rooms can account for a significant fraction. Thus, understanding the real environmental impact of medical procedures, surgical or otherwise, along a patient's care pathway, and implementing the mitigation measures we have discussed is part of a much bigger picture. It is one of the ways healthcare can help shift the trajectory of its carbon emissions and contribute to broader national and international decarbonization goals.

## Limitations

While we attempted to be comprehensive, usual limitations apply. We relied on available published data; some relevant studies may have been in progress or in gray literature (e.g., hospital internal reports) and not captured. We assumed the accuracy of each study's data and did not independently audit their calculations; differences in data quality or errors in original studies would propagate to our synthesis. We also could not meta-analyze quantitatively due to heterogeneity, instead using a narrative approach. Nonetheless, the consistency of themes across independent studies adds robustness to our conclusions.

A further important limitation is the exclusive focus of this review on carbon-equivalent emissions as the environmental endpoint. While greenhouse gas emissions currently represent the most pressing and most widely reported environmental impact of surgical care, other impact categories, including water consumption, eutrophication, toxicity, land use, and plastic and pharmaceutical pollution, are relevant and can diverge from the carbon signal. Mitigation actions optimized solely for carbon may therefore inadvertently shift environmental burdens to other impact categories (the so-called "burden-shifting" problem), for example when reusable devices with a lower carbon footprint entail higher water consumption or chemical emissions during reprocessing. Multi-criteria LCA frameworks and careful interpretation of trade-offs are needed to avoid unintended environmental consequences. As a minor additional limitation, the present review did not include product-level LCAs of medical devices, implants, or pharmaceuticals, which were outside its scope but could provide complementary insight into the upstream drivers of surgical emissions.

## Conclusion

Surgical operations contribute significantly to greenhouse gas emissions, but this impact can be dramatically reduced through targeted strategies. The 2023–2025 literature shows that by eliminating unnecessary single-use items, favouring reusable instruments and materials, choosing low-carbon anaesthetic techniques, optimizing operating room energy use, and rethinking care pathways to reduce travel and waste, the carbon footprint of surgery can be lowered by 30–80% in many cases without detriment to patient care. Surgeons, anaesthesiologists, nurses, and administrators all have roles to play: from micro-level decisions (e.g., using a reusable instrument set, turning off equipment, closing the lid on the anaesthetic vaporizer) to macro-level policies (investing in sustainable infrastructure, revising procurement contracts to prioritize low-carbon supplies, and institutionalizing telehealth where appropriate). Incorporating environmental sustainability into routine surgical practice is increasingly recognized as part of delivering high-quality care. Surgical departments can develop data-driven sustainability action plans, and researchers can focus on filling the remaining knowledge gaps, particularly in under-studied procedures and in standardizing assessment methods. Continued innovation, in device reusability, in energy-efficient technologies, and in clinical care models, will further empower the surgical community to reduce its carbon footprint. The studies reviewed provide clear proof that greener surgery is possible now, and the real challenge ahead lies in scaling up these solutions across healthcare systems globally, monitoring progress, and fostering a culture where the surgical team is as attuned to resource stewardship as it is to sterile technique or patient safety checklists. Sustainable surgery is not a distant ideal but an attainable goal.

## Supporting information

**S1 File. PRISMA 2020 Checklist.**
(DOCX)

**S2 File. Search strategy.**
(DOCX)

**S1 Table. Quality Assessment.**
(DOCX)

**S2 Table. Study inventory boundaries, data collected, and method used to calculate inventory results.**
(DOCX)

**S3 Table. Stated exclusions, assumptions and other limitations.**
(DOCX)

## Author contributions

**Conceptualization:** Beatrice Marchi.

**Data curation:** Anna Savio, Beatrice Marchi, Andrea Roletto.

**Formal analysis:** Anna Savio, Beatrice Marchi, Andrea Roletto.

**Investigation:** Anna Savio, Beatrice Marchi.

**Methodology:** Anna Savio, Beatrice Marchi.

**Supervision:** Pierangelo Guizzi, Giuseppe Milano, Lucio Enrico Zavanella, Simone Zanoni.

**Validation:** Andrea Roletto.

**Writing – original draft:** Anna Savio, Beatrice Marchi.

**Writing – review & editing:** Anna Savio, Beatrice Marchi, Andrea Roletto, Pierangelo Guizzi, Giuseppe Milano, Lucio Enrico Zavanella, Simone Zanoni.

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
