## [Decision Letter · Decision Letter 0]

3 Mar 2026

PONE-D-26-04683The Carbon Footprint of Surgical Operations: 2023-2025 Systematic Review UpdatePLOS One

Dear Dr. Savio,

Thank you for submitting your manuscript to PLOS ONE. After careful consideration, we feel that it has merit but does not fully meet PLOS ONE’s publication criteria as it currently stands. Therefore, we invite you to submit a revised version of the manuscript that addresses the points raised during the review process. Please submit your revised manuscript by Apr 17 2026 11:59PM. If you will need more time than this to complete your revisions, please reply to this message or contact the journal office at plosone@plos.org. Please include the following items when submitting your revised manuscript:

We look forward to receiving your revised manuscript.

Kind regards,

Manuel Herrador, Ph.D.

Academic Editor

PLOS One

Journal Requirements:

“This research received no external funding and no financial or non-financial support.”

Additional Editor Comments:

Dear authors,

Thanks for submitting your work to PLOS ONE.

In light of the reviewers’ suggestion for minor revisions, I concur with this recommendation based on the manuscript’s present stage of development.

Best regards

Reviewers' comments:

Reviewer's Responses to Questions

**Comments to the Author**

1. Is the manuscript technically sound, and do the data support the conclusions?

Reviewer #1: Yes

Reviewer #2: Yes

2. Has the statistical analysis been performed appropriately and rigorously? 

Reviewer #1: Yes

Reviewer #2: Yes

3. Have the authors made all data underlying the findings in their manuscript fully available?

Reviewer #1: Yes

Reviewer #2: Yes

4. Is the manuscript presented in an intelligible fashion and written in standard English?

Reviewer #1: Yes

Reviewer #2: Yes

5. Review Comments to the Author

Reviewer #1: The paper is an updated review of two earlier reviews on the carbon footprint of surgery. Given the rapid development of this field, the study is timely and appropriate. The authors identified 23 new studies in the last 3 years, which accounts for this.

The manuscript is well-written and of excellent quality. I agree with the interpretation of the studies and the authors highlight most relevant points. I also like the table with recommended mitigation strategies. The authors have set out five ambitious goals in the introduction. They deliver on most of these, but prehaps the scoping part of the review (goal 5) could be improved by clearly highlighting a research agenda on the identified knowledge gaps. My suggestion would be to use the two most extensive LCA's (ref 77 and 11) as a 'methodological reference'. It could be useful to interpet what these two papers did right and draw lessons from that for future studies. For instance, the inclusion of the whole care-pathway versus the inclusion of just the surgery itself.

On a minor level, the results section could be shortened by omitting some interpretations and sticking more closely to describing the data. Also, I believe fig 2 is of little added value and could be deleted. Lastly, the authods should add the sole focus on carbon as a major limitation. Although I agree that this is the most important environmental outcome at this point, reducing carbon should not lead to an increase in other environmental pollution outcomes (i.e. burden shifting). A minor limitation should be that this review did not include product LCAs.

I thank the authors for their laborious paper, it is of great value for the green OR community and I would love to see it published.

N.H. Sperna Weiland, Amsterdam, NL

Reviewer #2: Thank you for the opportunity to review this work. This is a well written and presented review.

Introduction: Whilst it is useful to detail the previous literature to justify the need for this review, I believe much of this could be moved to the discussion to make the introduction more concise. Your review aims are good but I believe e) could be made supplementary or an additional aim rather than a core finding.

Results: Well formatted results and interesting findings. I would consider adding a section of whether studies focussed solely on the perioperative phase versus the entire patient pathway and what boundaries were set for this in each study. We are becoming more and more aware that many of the emissions attributable to surgery occur befroe even entering the operating theatre and this shoud be commented on in your findings.

I am aware of many "eco-audits" for multiple specialties and in Plastic Surgery in particular I know many carbon footprinting studies have been performed. Is it possible your keywords search has missed these? If not, please can you clarify which of your exclusion criteria meant they were not included?

Discussion and Conclusion: Discussion generally sound. Would include some further comments on how the results of this study differ from the previous systematic reviews mentioned currently in your introduction. I would also include further discussion on how future carbon footpinting studies should be coducted to reduce hetergoneity and improve the quality to increase he utility of findings.

6. PLOS authors have the option to publish the peer review history of their article (what does this mean?). If published, this will include your full peer review and any attached files.

Reviewer #1: **Yes:**Nicolaas H. Sperna Weiland

Reviewer #2: **Yes:**Zahra Ahmed

---

## [Author Response · Author response to Decision Letter 1]

28 Apr 2026

Dear Reviewers,

We thank you for the careful and constructive review of our manuscript. The comments have substantially improved the quality, clarity, and rigour of the work, and we are very grateful for them. We have addressed every point raised, as detailed below.

All revisions are incorporated in the revised manuscript; section references in our responses follow the numbering of the revised version. The most substantive additions introduced in this revision are:

• Integration of three newly identified eligible studies on plastic and reconstructive surgery (Ahmed et al. 2024 on the DIEP flap pathway [25]; Ahmed et al. 2025a on the autologous microtia reconstruction pathway [30]; Ahmed et al. 2025b on the breast implant pathway [26), bringing the total from 23 to 26 included studies.

• A new sub-section in the Results addressing the perioperative-only versus whole-pathway divide as a second boundary dimension.

• Three new paragraphs in the Discussion: a comparison with the two prior systematic reviews, a methodological-standardisation paragraph, and a research-agenda paragraph that uses the two most comprehensive studies in our sample as methodological reference points.

• An expanded Limitations section that explicitly addresses the burden-shifting risk and the exclusion of product-level LCAs.

• Removal of the previous Figure 2 (study distribution by specialty); renumbering of the previous Figure 3 to Figure 2 with colour-coding by surgical specialty.

• Reframing of aim € as a supporting objective rather than a core finding.

Point-by-point responses follow below.

Reviewer #1

General appraisal: “The paper is an updated review of two earlier reviews on the carbon footprint of surgery. Given the rapid development of this field, the study is timely and appropriate. The authors identified 23 new studies in the last 3 years, which accounts for this. The manuscript is well-written and of excellent quality. I agree with the interpretation of the studies and the authors highlight most relevant points. I also like the table with recommended mitigation strategies.”

Response: We sincerely thank the Reviewer for this very kind and encouraging assessment, which is much appreciated. We have further strengthened the manuscript along the specific directions indicated below.

Comment 1.1: “The authors have set out five ambitious goals in the introduction. They deliver on most of these, but perhaps the scoping part of the review (goal 5) could be improved by clearly highlighting a research agenda on the identified knowledge gaps. My suggestion would be to use the two most extensive LCAs (ref 77 and 11) as a ‘methodological reference’. It could be useful to interpret what these two papers did right and draw lessons from that for future studies. For instance, the inclusion of the whole care-pathway versus the inclusion of just the surgery itself.”

Response: We thank the Reviewer for this constructive and very specific suggestion. We have added a dedicated paragraph in the Discussion that develops the scoping component into an explicit research agenda. As suggested, we use the two most comprehensive assessments in our sample as methodological reference points: the CABG study by van Bree et al. [11] and the minimally invasive gynaecological pathway assessment by Cohen et al. [36]. (Reference [77] in the version originally reviewed corresponds to reference [36] in the revised manuscript following renumbering after the addition of the three plastic-surgery studies; the substance of the Reviewer’s suggestion has been preserved verbatim.) We describe what these two studies did particularly well and we draw five priorities for future research from this approach:

• extending whole-pathway assessments to high-volume procedures that remain poorly characterized (e.g. appendectomy, cholecystectomy, neurosurgical and paediatric operations);

• developing comparative LCAs of robotic versus conventional surgery to quantify the carbon trade-offs of new technologies;

• producing product-level LCAs of medical devices, implants and pharmaceuticals to complement pathway-level assessments;

• integrating carbon metrics with clinical outcomes (e.g. emissions per QALY) to enable value-based sustainability analyses;

• longitudinal re-auditing of pathways after sustainability interventions, to verify effectiveness under routine conditions.

The whole-care-pathway versus theatre-only distinction explicitly raised by the Reviewer is now also the subject of a dedicated Results sub-section.

Location of change: Discussion, new research-agenda paragraph; Results, new paragrapg

Comment 1.2: “On a minor level, the results section could be shortened by omitting some interpretations and sticking more closely to describing the data.”

Response: Agreed. We have reviewed the Results section and tightened wording in several passages where interpretive content had been embedded with descriptive findings. Explanatory and interpretive material has either been trimmed or moved to the Discussion; data-focused reporting has been preserved throughout.

Location of change: Minor edits distributed across Results section.

Comment 1.3: “Also, I believe fig 2 is of little added value and could be deleted.”

Response: We agree. The previous Figure 2 (“Study Distribution Across Surgical Specialties”) has been removed. The previous Figure 3 (range of reported carbon footprints) has been renumbered to Figure 2 and has been redrawn with colour-coding by surgical specialty, so that the specialty information previously carried by the deleted figure is now integrated into the retained one. All in-text references to the figures and the figure caption have been updated accordingly.

Location of change: Results §3.1 (in-text references updated); Figure 2 caption rewritten; new figure file with colour-coded specialties provided.

Comment 1.4: “Lastly, the authors should add the sole focus on carbon as a major limitation. Although I agree that this is the most important environmental outcome at this point, reducing carbon should not lead to an increase in other environmental pollution outcomes (i.e. burden shifting).”

Response: We thank the Reviewer for raising this fundamental point. We have added a new paragraph in the Limitations section that explicitly acknowledges:

• that the present review focuses exclusively on greenhouse-gas emissions as the environmental endpoint;

• that other impact categories (water consumption, eutrophication, toxicity, land use, plastic and pharmaceutical pollution) are relevant and can diverge from the carbon signal;

• that mitigation actions optimised solely for carbon may inadvertently shift environmental burdens to other impact categories (the so-called “burden-shifting” problem ) for example when reusable devices with a lower carbon footprint entail higher water consumption or chemical emissions during reprocessing;

• that multi-criteria LCA frameworks and careful interpretation of trade-offs are needed to avoid unintended environmental consequences.

Location of change: Limitations, new paragraph added.

Comment 1.5: “A minor limitation should be that this review did not include product LCAs.”

Response: Agreed. In the same new paragraph of the Limitations section we have explicitly noted that the present review did not include product-level LCAs of medical devices, implants or pharmaceuticals, and that these were outside its scope but could provide complementary insight into the upstream drivers of surgical emissions. This is also reflected in the research-agenda paragraph in the Discussion, where product-level LCAs are listed among the priority areas for future research.

Location of change: Limitations, same new paragraph; Discussion, research-agenda paragraph.

Closing comment: “I thank the authors for their laborious paper, it is of great value for the green OR community and I would love to see it published.”

Response: We are very grateful to the Reviewer for the kind closing remarks and for a review that has been both rigorous and genuinely constructive.

Reviewer #2

General appraisal: “Thank you for the opportunity to review this work. This is a well written and presented review.”

Response: We sincerely thank the Reviewer for this positive overall assessment and for the specific, expert suggestions that follow, which have allowed us to materially strengthen the manuscript.

Comment 2.1: “Introduction: Whilst it is useful to detail the previous literature to justify the need for this review, I believe much of this could be moved to the discussion to make the introduction more concise.”

Response: Agreed. We have shortened the Introduction’s treatment of the two prior systematic reviews [6,7] to a concise summary of their scope and study counts. The detailed comparison of the present findings with those prior reviews is now presented in a dedicated new paragraph at the start of the Discussion.

Location of change: Introduction; Discussion, new first paragraph.

Comment 2.2: “Your review aims are good but I believe e) could be made supplementary or an additional aim rather than a core finding.”

Response: Agreed and addressed. The previous five-aim list has been condensed into four core aims (quantitative summary; emissions hotspots; mitigation strategies; methodological variability), and the previous aim (e) has been reframed as a supporting objective in the closing paragraph of the Introduction (“identifying knowledge gaps and research needs for the field of sustainable surgery”), with its full development now placed in the Discussion as a research-agenda paragraph. This balances Reviewer #2’s suggestion with Reviewer #1’s request to strengthen the scoping component.

Location of change: Introduction; Discussion, research-agenda paragraph.

Comment 2.3: “Results: Well formatted results and interesting findings. I would consider adding a section of whether studies focussed solely on the perioperative phase versus the entire patient pathway and what boundaries were set for this in each study. We are becoming more and more aware that many of the emissions attributable to surgery occur before even entering the operating theatre and this should be commented on in your findings.”

Response: We fully agree and thank the Reviewer for this important suggestion. The system-boundary section has been restructured around two distinct dimensions: (a) the breadth of GHG scopes considered (Scope 1/2/3 coverage), and (b) the temporal extent of the patient pathway included (perioperative-only versus whole-pathway). The second dimension is now developed in a dedicated paragraph that:

• defines the methodological divide and its implications for the magnitude and the composition of the reported footprint;

• lists narrow-boundary studies focused on operating-theatre activity (e.g. [21,27]) alongside whole-pathway assessments (e.g. [10,11,18,25,26,30,36]);

• concludes that whole-pathway assessments are more representative of the true climate burden of surgery, while perioperative-only audits are better suited to evaluating intraoperative interventions, and recommends that future studies explicitly state their system boundaries.

This same boundary perspective also anchors a further illustration in the quality-assessment section, where we use the three Ahmed et al. studies as an intra-team, intra-centre, intra-method comparison showing that boundary choices alone can drive an order-of-magnitude difference in the reported footprint (1,005 vs 234 vs 49 kgCO₂e for the microtia, DIEP and breast implant pathways respectively).

Location of change: Results, new sub-section, second paragraph dedicated to the perioperative vs whole-pathway dimension.

Comment 2.4:“I am aware of many ‘eco-audits’ for multiple specialties and in Plastic Surgery in particular I know many carbon footprinting studies have been performed. Is it possible your keywords search has missed these? If not, please can you clarify which of your exclusion criteria meant they were not included?”

Response: We are very grateful to the Reviewer for raising this point. We re-ran the literature search and retrieved three eligible studies in plastic and reconstructive surgery that met our inclusion criteria that we initially excluded:

• [25] Ahmed Z et al. Sustainability in Reconstructive Breast Surgery: An Eco-audit of the Deep Inferior Epigastric Perforator Flap Pathway. Plast Reconstr Surg Glob Open. 2024;12: e6374.

• [26] Ahmed Z et al. Sustainability in breast surgery: An eco-audit of the breast implant pathway. J Plast Reconstr Aesthet Surg. 2025;105: 219–229.

• [30] Ahmed Z et al. Evaluating the Environmental Impact of Quaternary Plastic Surgery: An Eco-audit of the Autologous Microtia Reconstruction Pathway. Plast Reconstr Surg Glob Open.

All three studies have now been fully integrated into the review. We are also grateful to the Reviewer for their important scientific contribution to this field and for prompting this substantive expansion of the evidence base covered by the review.

Location of change: Abstract; Methods; PRISMA flow and Figure 1; Results; Discussion; References [25], [26], [30] (citation numbering propagated throughout); Table 1; Table 2; S1 Table; S2 Table; S3 Table.

Comment 2.5:“Discussion and Conclusion: Discussion generally sound. Would include some further comments on how the results of this study differ from the previous systematic reviews mentioned currently in your introduction.”

Response: Agreed. We have added a new first paragraph to the Discussion that directly compares the present 26-study corpus with the 8 studies covered by Rizan et al. [6] and the 7 studies covered by Robinson et al. [7]. The paragraph quantifies the expansion of the evidence base, summarizes the broader specialty coverage and more diverse geographical distribution, notes the increased uptake of formal LCA frameworks (ISO 14040/14044, GHG Protocol) in the recent literature, and identifies two findings that extend the prior syntheses: the prominence of travel and anaesthetic gases as major contributors in studies with broader system boundaries (only marginally addressed in [6,7]), and the previously uncaptured upper end of the footprint range revealed by whole-pathway plastic-surgery.

Location of change: Discussion, new first paragraph.

Comment 2.6:“I would also include further discussion on how future carbon footprinting studies should be conducted to reduce heterogeneity and improve the quality to increase the utility of findings.”

Response: We have addressed this point with a dedicated new paragraph in the Discussion that proposes convergence on a small set of shared methodological principles for future carbon-footprinting studies in surgery: explicit adoption of a recognized LCA framework (ISO 14040/14044 or the GHG Protocol Product Standard); transparent reporting of system boundaries, functional units and data sources; clear declaration of included and excluded life-cycle stages; systematic uncertainty analysis; and a preference for whole-pathway assessments over perioperative-only audits whenever the clinical context allows. We also recommend reporting checklists tailored to healthcare LCAs and open-access repositories of healthcare-specific emission factors as practical tools to improve consistency and enable cumulative evidence synthesis.

Location of change: Discussion, new methodological-standardisation paragraph.

We thank the Reviewers again for their time and constructive input, and the Editor for the opportunity to revise the manuscript. We believe the revisions have substantially strengthened the work, and we look forward to your decision.

Kind regards,

Anna Savio

Corresponding author, on behalf of all co-authors

---

## [Editor Report · Decision Letter 1]

29 Apr 2026

The Carbon Footprint of Surgical Operations: 2023-2025 Systematic Review Update

PONE-D-26-04683R1

Dear Dr. Savio,

We’re pleased to inform you that your manuscript has been judged scientifically suitable for publication and will be formally accepted for publication once it meets all outstanding technical requirements.

Kind regards,

Manuel Herrador, Ph.D.

Academic Editor

PLOS One

Additional Editor Comments (optional):

Dear Authors,

We are pleased to inform you that your manuscript has been accepted for publication in PLOS ONE.

You have successfully addressed all previous feedback, and the reviewers had no further comments on your revised manuscript. Thank you for your hard work on these revisions and for submitting your valuable research to our journal.

You will receive further instructions from the production team shortly regarding the next steps for publication.

Sincerely
---

## [Editor Report · Acceptance letter]

PONE-D-26-04683R1

PLOS One

Dear Dr. Savio,

I'm pleased to inform you that your manuscript has been deemed suitable for publication in PLOS One. Congratulations! Your manuscript is now being handed over to our production team.

Kind regards,

on behalf of

Dr. Manuel Herrador

Academic Editor

PLOS One